# Prevalence of HIV seropositive status and associated factors among family members of index cases of antiretroviral clinical attendants in Sodo Town, Southern Ethiopia

**Alemayehu Kefale[1]\*, Kassa Daka[2], Amene Abebe[2], Dereje Haile[2]\*, Kebreab Paulos[2], Abdulbasit Sherfa[3], Animut Addis[4], Muluken Gunta[1], Asaminew Ayza[1], Jegnaw Wolde[1]**

1 Wolaita Zone Health Department, Wolaita Sodo, Southern Ethiopia, 2 School of Public Health, Wolaita Sodo University, Sodo, Southern Ethiopia, 3 Department of Public Health, Werabe University, Werabe, Southern Ethiopia, 4 Department of Disease Prevention and Control Core Process, ICAP, Hawassa, Ethiopia

\* alemayehukefale@yahoo.com (AK); derehaile2010@gmail.com (DH)

**Data Availability Statement:** All relevant data are within the manuscript and its Supporting Information files.

## Abstract

### Background

Human immunodeficiency virus is primarily transmitted through sexual contact with an infected partner and babies born to mothers infected with the virus. Partners of people living with HIV and children whose parents have HIV are at higher risk of contracting HIV unless they take preventive measures. This study aimed at identifying prevalence and determinants of HIV infection among family members of index cases on antiretroviral treatment (ART).

### Methods

A community-based cross-sectional study was conducted among 623 randomly selected family members of HIV index cases in Sodo Town from February to June 2021. A pre-tested structural questionnaire was used to collect data. Binary logistic regression was used to identify variables independently associated with the outcome variable. The adjusted odds ratio (AOR) with 95% confidence interval (CI) was used to show the strength of association, and a $P$-value 0.05 was used as a cut-off point to determine the level of statistical significance of point estimate.

### Results

This study revealed that 31.5% (95%CI: 27.6–35.2%) of family members of index cases were HIV seropositive. In subgroup analysis, this study also revealed that 11.1% (95%CI 8.4–14.5%) of biological children and 69.6% (95%CI 63.1–75.6%) of spousal partners of index cases were HIV seropositive. Immediate ART initiation of index cases (AOR = 0.148, 95%CI: 0.067–0.325), being bedridden or ambulatory functional status at enrollment (AOR = 7.71, 95%CI: 3.5–17), and baseline CD4 level of 350 cells/ml (AOR = 8.06, 95%CI: 1.8–36) were statistically significant with the outcome variable among biological children. Among

**Funding:** The author(s) received no specific funding for this work.

**Competing interests:** The authors have declared that no competing interests exist.

spousal partners, STI history or symptoms (AOR = 5.7, 95%CI: 1.86–17.5), early disclosure (AOR = 0.062, 95%CI: 0.024–0.159), immediate ART initiation (AOR = 0.172, 95%CI: 0.044–0.675), and duration of infection (AOR = 5.09, 95%CI: 1.8–14.4) were statistically associated with the outcome variable.

## Conclusion

As evidenced by our data, the risk of HIV among family members of index cases is high. Interventions like immediate ART initiation, early disclosure, screening, and early treatment of STIs for minimizing HIV transmission might be given.

## Introduction

Index testing is a voluntary process where counsellors or health care providers ask a newly diagnosed HIV positive patient or an HIV positive patient already accessing HIV treatment to list all partners who could have been exposed to HIV through them, the index cases [1].

ICT has two models of testing(1)Facility Based ICT that a counselor talks with the index cases on ART at the health facility, and gets the names and contact information for sexual partner(s) and biological children under 15 years old in Antiretroviral Treatment (ART) service giving facility and (2) community-based ICT: a counsellor traces sexual partner(s) of index cases and their biological children who are under 15 years of age through home-to-home visits, offers HTS, and links newly identified People Living with HIV (PLHIV) to ART services [1].

According to the United Nations Program on HIV/AIDS (UNAIDS), 37.7 million people worldwide were living with HIV/AIDS in 2021, including 1.7 million children under the age of 15 years [2]. In Eastern and Southern Africa (ESA), 20.6 million PLHIV were living in 2021, and of which 670,000 people were newly infected by HIV [2]. Despite this burden, a transformative and remarkable progress has been made in the past decades with regards to the expansion of ART indicating that from of the total number of PLHIVs in ESA in 2021, 16 million were on treatment [2]. In 2021, in Ethiopia, there were an estimated 620,000 PLHIV, of which 44,000 (7.1%) were children under 15 years of age and 360,000 (62%) were females [3].

There were 12,000 new annual infections, among which 2,800 were children under 15 years of age. There were 13,000 annual deaths (11,000) adults and 2,000 children under 15 years of age. Of the total PLHIV in Ethiopia, 483,127 (465,457 adults and 17,670 children under 15 years of age) were receiving ART [3]. According to the Ethiopian Demographic and Health Survey (EDHS) 2016, the national HIV prevalence in adults aged 15–49 years was 0.9% (0.7% in males and 1.2% in females) [4].

'HIV/AIDS in Ethiopia is regularly categorized as "generalized"among the adult population, with a significant variation in the burden of HIV across a geographic area and population subgroups [5]. The urban HIV prevalence is seven times higher than the rural HIV prevalence. Among the regional states, the HIV prevalence ranges from 4.8% in Gambella to below 0.1 in Ethiopian Somali region [6]. There is also a disproportionally high HIV burden among population groups such as sexual partners of PLHIV, female commercial sex workers, long truck drivers, prisoners, widowed and divorced urban women, and mobile workers, where the HIV prevalence rate is 28%, 23%, 4.9%, 4.2%, 3.5%, and 1.5%, respectively [4, 7].

Ethiopia has changed its universal HIV testing strategy, which is testing all individuals in the population who were at risk to yield-based target test and treat strategy. Family members of index cases as a target population for this strategy [8]. For the implementation, the country has adopted the World Health Organization (WHO) Partner and Family-based Index Case

Testing (PFBICT) or ICT strategy; a voluntary process where counselors and/or healthcare workers ask index cases to list all sexual partners in the past year and under15-year-old biological children. This breaks the chain of transmission by offering HTS to people who are exposed to HIV to provide preventive services for those testing HIV-negative, and if positive, link them to care and treatment services, a cascade that will contribute to attaining the 95-95-95 targets. Some index cases might not have yet disclosed their HIV status, exposing the family members to an ongoing risk of contracting HIV. Therefore, index cases' concerns should be addressed to improve disclosure and testing service uptake among sexual partners and under 15-years old biological children [1]. In Ethiopia, the majority of HIV infections among spousal partners are heterosexual, and 90% of HIV transmission among biological children under the age of 15 is through mother-to-child transmission (MTCT) [4, 9].

This indicates that HIV-negative spousal partners of index cases and biological children under 15 years of age whose parents have HIV or family members of index cases are at a higher risk of contracting HIV. To reduce the risk of HIV transmission, family members of index cases would need to know their HIV status. Testing for HIV and linking those who are positive to care and treatment services will improve the health outcomes of the seropositive family members [7].

Nevertheless a study conducted on the assessment of tracing family members of HIV- positive people for HIV testing and associated factors among ART clinic attendants at a referral hospital in Northwest Ethiopia revealed that many children and adults were traced and tested for HIV. Yet, there were many untested partners, children and other individuals living with index cases who were not tested [10].

However, no similar study was conducted to assess HIV seropositive status and associated factors among family members of index cases and ART clinic attendants in this town. This study aimed to assess the prevalence of seropositive status and associated factors among family members of index cases among ART clinic attendants in Sodo Town. 'HIV serostatus information on family members of index cases, which will contribute to improving HIV care and testament services and epidemic control in Ethiopia'.

## Materials and methods

### Study area and period

The study was conducted at Sodo Town, which is located in Wolaita Zone, South Ethiopia. The Town, which is the capital city of the Zone, is located 380 km south of Addis Ababa, the capital city of Ethiopia, and 157 km south of Hawassa, the capital city of Southern Nations, Nationalities, and Peoples Region (SNNPR). Sodo Town is further divided into three sub-cities and 25 kebele (lowest administrative entity)".

According to the 2021 population and housing census projection, Sodo Town has a total population of 244,817, of which 121,184 are males and 123,633 are females. There are two hospitals, three health centers, and 25 health posts. There are 3 ART service-delivery facilities in the town, of which two are hospitals and the remaining one is a health center. This study was conducted from February 18–June 30, 2021.

### Study design

The study was a community-based cross-sectional study.

### Source population

The source population of this study seems all patients coming to the health facilities involved in this study.

## Study population

All family members aged 2–14 years old, biological children and spousal partners of index cases of ART clinic attendants in Sodo Town.

## Study participants

The study subjects seem all index cases coming at ART clinics at the study health facilities and their family members.

## Inclusion and exclusion criteria

**Inclusion criteria.**   An index case living in Sodo who at least had one biological child aged 2–14 years or a spousal partner.

Biological children 2–14 years old and an index case living in Sodo who at least had one biological child aged 2–14 years or a spousal partner.

**Exclusion criteria.**   Index cases who didn't disclose their status

Biological children (2–14 years old) and spousal partners who couldn't respond due to serious illness or not willing to participate

## Sample size determination

In this study, a sample size was determined using the single population proportion formula by assuming a proportion of 50%, a margin of error of 5%, a 95% confidence interval, and a non-response rate of 10%. The calculated sample size was 423. However, a total of 651 eligible index cases of HIV were identified from the registry, so we used the census method to recruit all study participants.

## Sampling technique and procedure

The ART clinics at all three health facilities in Sodo Town (Sodo Health, Sodo Teaching and Referral Hospital, and Sodo Christian General Hospital) were included as study sites. According to the report of each facility, the total HIV-positive patients currently on ART at the end of September 2020 was 2,317 (Sodo HC = 662, Sodo Teaching and Referral Hospital = 1530, and Sodo Christian General Hospital = 125). And among these 2,213 were adults aged $\geq$ 18 years. As a first step from the PFBICT registration book, we identified all adult clients living in Sodo Town, which were 1,144. By using HIV testing positive dates between each spousal partner, we then identified 834index cases. Of the total index case (834), only 723 who had at least one family member were selected and given a serial code. The unique ART code numbers given by the national program to the facilities were used to identify each study facility." In a study done at public health facilities in Butajira Town, Southern Ethiopia, HIV-positive status disclosure among HIV-positive adult patients attending ART clinics, the magnitude of non-disclosure was 10% [11].

Of 723 index patients, 72 were expected not to disclose and 651 were expected to disclose their status to their family members. Since all index cases were manageable, we included them all and selected them as total index cases. Finally, from each disclosed index case, only one family member was selected by lottery method, as a result, had a total of 651 family members from all ART sites.

## Study variables

**Dependent variable.**   HIV seropositive status among biological children of index cases

HIV seropositive status among spousal partners of index cases

**Independent variables.**   Socio-demographic and economic variables: age, sex, residence, marital status, education, wealth status, and occupation.

Index case-related factors: duration of a relationship, ART initiation status, enrollment status, history of treatment failure, history of adherence, WHO stage, viral load (VL), CD4+ and disclosure status.

Sexual and substance use behavior of spousal partners: condom use, substance use (alcohol drinking, chat chewing &tobacco smoking), number of lifetime sexual partnership or multiple sexual partners, and type of current marriage (polygamous/monogamous).

HIV testing and STIs history: ever tested or not, sexually transmitted infections.

## Data collection instrument and procedure

The data collection tool was an interviewer-administrator structured questioner developed through the review of related literature. It contains information on socio-demographic and economic factors, sexual and substance use behaviors of spousal partners, HIV testing and STIs-related history factors, and index case-related factors.

Following disclosure of HIV status by the index cases, data collectors fixed appointments with them and their families for home visits At the start of the visit, the purpose of the study was explained to study participants and informed verbal consent obtained from spousal partners and consent for all biological children was obtained from the parents, and assent was obtained from children 12–14 years of age. In those who were HIV-positive, only interviews were administered by the investigators, but if their status was unknown or negative, the participants were interviewed and tested for HIV. Parents were interviewed on behalf of biological children. The testing procedure strictly followed the current national testing algorithm. Unique ART numbers were used to link primary index case data with that of the family members and with their secondary data in patients' records for analysis.

## Operational definitions

Index case: first diagnosed spousal partner with HIV infection in the family

Biological children: children born from the index case and aged 2–14 years

Spousal partner: individuals who are currently married or engaged or in a steady relationship

Family members of index case: biological children and/or spousal partner

HIV seropositive status: being reactive as per the national HIV testing algorithm test

Treatment failure: clinically stage III or IV, CD4+ <200cells/ml or viral load >/1000 copies/ml after 6 months of ART treatment

Disclosure: Early, when HIV positive family members share their status to his family members within one month, and late, if not disclosed within a month or disclosed after a month of diagnosis

Condom use: always, if used at each sexual intercourse, sometimes, if not used at every sexual intercourse e.g. for only prevention of pregnancy and never, not used at all

ART initiation status: Immediate, if clients started ART whenever eligible, and late, if not started ART though eligible to start based on eligibility criteria.

Viral load status: suppressed, if number of copies /ml <1000, and unsuppressed, if number of copies /ml>/1000 after taking ART at least for 6.months in the last 12 months.

## Data management and quality control

The English version of the questionnaire was translated into the Amharic language and checked for comprehension and consistency. Then it was translated back into English. The Amharic version was used for to collect data. The questionnaire was pre-tested by the data collectors and the supervisors in 5% of the sample at Boditi Health Center. The results were used to modify the questionnaire as appropriate. Eight data collectors and two supervisors working in the ART clinic

were recruited and trained. Two-day training introduced the objective of the study, familiarized data collectors with the questionnaire, the sampling procedure and interviewing techniques, and the HIV testing procedures. The supervisors and the principal investigator had a close follow-up of the day-to-day data collection process to ensure completeness and consistency of the questionnaire administered each day. The collected data were rechecked for its completeness and consistency by the supervisors and principal investigators before being transferred into computer software. A non-overlapping numerical code was given for each questionnaire.

## Data analysis procedure

EPI data 4.2 statistical software was used for data entry, cleaning, and codingof the variables. The data were exported to SPSS version 25 for analysis. Descriptive statistics were used to summarize the data and check for the distribution of the study variables. A bivariate analysis was done by using a binary logistic regression. A$P$-value of 0.25 was used as a cut-off point to select candidate variables for multivariate logistic regression, the analysis used to determine the association between the independent and dependent variables having controlled for confounding effects. AOR with 95%CI was used to determine the level association. The Hosmer and Lemeshow's goodness of fit test was used to check for model fitness. A variance inflation factor (VIF) or tolerance test was done to check for correlation. A $P$-value of 0.05 was used as a cut-off pointto determine statistical significance and the results were presented in narratives and tables.".

## Ethical consideration

Ethical clearance was obtained from the Ethical Review Board of Wolaita Sodo University College of Health Science and Medicine. Formal permission letters for each health facility were written by the Zonal Health Department and Town Health Office. The purpose of the study was explained to the study participants. Both informed written and verbal consent were obtained from the parents. Moreover, assent was obtained from the parents for the children aged 12–14 years.

The information obtained from the study participants was kept confidential. The HIV testing and interview were undertaken privately in a scheduled area. Family members who tested negative were counseled about further preventive measures, and those who were positives were linked to a nearby ART clinic where care and support were given according to the national guidelines.

## Results

### Socio-demographic and economic characteristics of family members of index cases

A total of 623 family members of index cases in Sodo Town were Interviewed, which was a response rate of 94%. While the majority, 406 (65.2%), were biological children, nearly one-third 217(34.8%) were spousal partners. The mean (SD) age of biological children was 8.9 (± 3.66) and of the spousal partner was 36.9 (± 9.26) years. Of the respondents, nearly half (51.5%) were females, more than half (54.6%) were Orthodox Christians, and one-third (32.6%) of the respondents wealth status in the first quartile. Including 1072 biological children (2–14) years old (1.72 biological children per index case), a total of 2932 family members (4.71 family members per index case) were identified from 623 index cases (Table 1).

Even though we expected 651 index cases disclosures from 723 eligible index cases to determine sample size, 662 were disclosed during data collection and used as the denominator to determine non-response rate.

**Table 1. Socio-demographic and economic characteristics of members of index cases in Sodo Town, Southern Ethiopia, 2021 (N = 623).**

| Variables | Category | Frequency | Percentage |
|---|---|---|---|
| Sex | Male | 302 | 48.5 |
| | Female | 321 | 51.5 |
| Age | <5 | 87 | 14.0 |
| | 5–9 | 126 | 20.2 |
| | 10–14 | 192 | 30.8 |
| | 15–29 | 43 | 6.9 |
| | ≥30 | 175 | 28.1 |
| Religion | Orthodox | 340 | 54.6 |
| | Protestant | 249 | 40.0 |
| | Muslim | 26 | 4.2 |
| | Catholic | 8 | 1.3 |
| Wealth index | First quartiles | 203 | 32.6 |
| | Second quartile | 211 | 33.9 |
| | Third quartile | 209 | 33.5 |
| Family size | 1–3 | 129 | 20.7 |
| | 4–6 | 447 | 71.7 |
| | ≥7 | 47 | 7.5 |
| 2–14 yrs. old biological child size | <3 | 506 | 81.2 |
| | ≥3 | 117 | 18.8 |
| Alive parents | Father | 26 | 4.2 |
| | Mother | 123 | 19.7 |
| | Both | 474 | 76.1 |
| Relation to the index | Spouse | 217 | 34.8 |
| | Biological child | 406 | 65.2 |

## Sexual and substance use behavior of spousal partners of index case

Of the 217 spousal partners, nearly one-fourth (23%) had 10 or more lifetime sexual partners. The majorities (84.8%) were in a monogamous relationship; 69.1% had been living together with their spouse for more than ten years; and (68.2%) had an age difference of ten years or more between spousal partners. Despite the fact that most (76.5%) had been educated on condom use, more than half (60.8%) had never used one during sexual intercourse with their spousal partners. While the majority (78.8%) had never smoked tobacco, 70% had chewed chat, 9.7% had drunk alcohol daily (Table 2).

## HIV testing history and serostatus of family members of index cases

Of the 267 family members, most (90.9%) had ever been tested for HIV, and 57 (9.1%) had never been tested. Of the ever-tested participants, 186 (32.9%) had positive status. Of those who had never been tested or whose results were negative (437), 10 (2.8%) were newly identified positives. The overall prevalence among family members was 31.5% (196/623 cases) (Table 3).

In a sub-group analysis of the biological children (N = 406), 353 (86.9%) had ever been tested, and 53 (13.1%) have not been tested. Of the ever-tested participants, 37 (10.5%) had positive status. Of those who had never been tested or whose results were negative, eight (2.8%) were newly identified positives. Of all the biological child respondents, 45 (11.1%) were positive and 361 (88.9%) were negative (Table 4).

**Table 2. Sexual and substance use behavior of spousal partners of index cases in Sodo Town, Southern Ethiopia, 2021 (N = 217).**

| Variables | Category | Frequency | Percentage |
|---|---|---|---|
| Number of life time sexual partners | <5 | 89 | 41.0 |
| | 5–9 | 78 | 35.9 |
| | ≥10 | 50 | 23.0 |
| How do you describe the type of your current marriage | Polygamous | 33 | 15.2 |
| | Monogamous | 184 | 84.8 |
| Is this your first marriage or partnership | Yes | 28 | 12.9 |
| | No | 189 | 87.1 |
| Are you the first spouse for your spouse | Yes | 17 | 7.8 |
| | No | 200 | 92.2 |
| Duration of spousal partners living together in year | <5 | 26 | 12.0 |
| | 5–9 | 41 | 18.9 |
| | ≥10 | 150 | 69.1 |
| Age difference between spousal partners | <5 | 22 | 10.1 |
| | 5–9 | 47 | 21.7 |
| | ≥10 | 148 | 68.2 |
| Ever discussed condom use with your spousal partner | Yes | 166 | 76.5 |
| | No | 51 | 23.5 |
| Ever used a condom during sexual intercourse with your spousal partner | Yes | 98 | 45.2 |
| | No | 119 | 54.8 |
| How regularly did you use condoms during sexual intercourse with spousal partner | Never | 132 | 60.8 |
| | Sometimes | 72 | 33.2 |
| | Always | 13 | 6.0 |
| Ever drunk alcohol | Never | 42 | 19.4 |
| | Occasionally | 154 | 71.0 |
| | Daily | 21 | 9.7 |
| Ever used Tobacco | Never | 171 | 78.8 |
| | Occasionally | 26 | 12.0 |
| | Daily | 20 | 9.2 |
| Ever chaw chat | Never | 152 | 70.0 |
| | Occasionally | 65 | 30.0 |

In a sub-group analysis of the spousal partners (N = 217) nearly one-fourth (23%) had STI symptoms or a history of STI in the 12 months prior to data collection. All (100%) heard about HIV testing and counseling service provision and thought testing would be important to family members of PLHIVs. A majority (98.2%) of them were ever tested, and 149 (70%) had HIV

**Table 3. HIV serostatus and testing history-related factors of family members of index cases in Sodo Town, Southern Ethiopia, 2021.**

| Variables | Category | Frequency | Percentage |
|---|---|---|---|
| Have you ever tested for HIV(N = 623) | Yes | 566 | 90.9 |
| | No | 57 | 9.1 |
| What was the result if you had been tested(N = 566) | Positive | 186 | 32.9 |
| | Negative | 380 | 67.1 |
| If never been tested or the result was negative, what is the current result? (practically confirmed)(N = 437) | Positive | 10 | 2.29 |
| | Negative | 427 | 97.71 |
| Final current HIV serostatus (N = 623) | Positive | 196 | 31.5 |
| | Negative | 427 | 68.5 |

**Table 4. HIV serostatus and testing history of biological children of index cases, Sodo Town, Southern Ethiopia, 2021 (N = 406).**

| Variables | Category | Frequency | Percentage |
|---|---|---|---|
| Have you ever tested for HIV | Yes | 353 | 86.9 |
| | No | 53 | 13.1 |
| What was the result if you had been tested(n = 353) | Positive | 37 | 10.5 |
| | Negative | 316 | 89.5 |
| If never been tested or the result was negative, what is the current result? (practically confirmed)(n = 369) | Positive | 8 | 2.17 |
| | Negative | 361 | 97.83 |
| Final current HIV serostatus | Positive | 45 | 11.1 |
| | Negative | 361 | 88.9 |

positive status. Of those who had never been tested or whose results were negative, two (3%) were newly identified HIV positive patients. Of the spousal partners respondents, 151 (69.6%) were HIV positive and 66 (30.4%) were negative (Table 5).

## Characteristics of index cases of family members

From a total of 623 index cases, 386 (62%) disclosed their status to at least one family member within a month of diagnosis, and the majority of 476 (76.4%) started ART immediately as they were eligible. Most, 97% and 79.9%, of the index cases had current and at enrollment working functional status, respectively. Five hundred fifty eight (89.6%) had a current suppressed viral load and 330 (53%) had a low CD4 count (350 cells/ml) at ART enrollment. The majority, 505 (81.1%) and 406 (65.2%) of the index cases had no history of treatment failure and poor adherence, respectively (Table 6).

## Factors associated with HIV seropositive status among biological children

The bivariate binary logistic regression found five variables that affected the HIV seropositive status of the family members of index cases of HIV. These were immediate ART initiation if eligible; functional status at enrollment; WHO stage at enrollment; duration of infection since known; and baseline CD4 of index cases of HIV (Table 7).

**Table 5. Sexually transmitted infections and HIV testing history of spousal partners of index cases in Sodo Town, Southern Ethiopia, 2021 (N = 217).**

| Variables | Category | Frequency | Percentage |
|---|---|---|---|
| Had history or symptoms of STIs (N = 217) | Yes | 50 | 23.0 |
| | No | 167 | 77.0 |
| Heard about HIV testing and counseling service provision to family members PLHIVs(N = 217) | Yes | 217 | 100 |
| | No | 0 | 0 |
| Do you think testing is important to family members of PLHIVs (N = 217)? | Yes | 217 | 100 |
| | No | 0 | 0 |
| Have you ever tested for HIV(N = 217)? | Yes | 213 | 98.2 |
| | No | 4 | 1.8 |
| What was the result if you had been tested (N = 213)? | Positive | 149 | 70.0 |
| | Negative | 64 | 30.0 |
| If never been tested or the result was negative, what is the current result? (practically confirmed) (N = 68) | Positive | 2 | 2.94 |
| | Negative | 66 | 97.06 |
| Final current HIV serostatus (N = 217) | Positive | 151 | 69.6 |
| | Negative | 66 | 30.4 |

**Table 6. Clinical and behavioral characteristics of HIV index cases in Sodo town, Southern Ethiopia, 2021 (N = 623).**

| Variables | Category | Frequency | Percentage |
|---|---|---|---|
| Notification of disclosure of HIV status | ≥1month | 237 | 38.0 |
| | <1month | 386 | 62.0 |
| Immediate ART initiation | Yes | 476 | 76.4 |
| | No | 147 | 23.6 |
| Functional status at enrollment | Bedridden | 40 | 6.4 |
| | Ambulatory | 85 | 13.6 |
| | Working | 498 | 79.9 |
| Current functional status | Bedridden | 5 | 0.8 |
| | Ambulatory | 14 | 2.2 |
| | Working | 604 | 97 |
| WHO stage at enrollment | Stage One | 265 | 42.5 |
| | Stage Two | 167 | 26.8 |
| | stage Three | 152 | 24.4 |
| | stage Four | 39 | 6.3 |
| Current WHO stage | Stage One | 593 | 95.2 |
| | Stage Two | 25 | 4.0 |
| | Stage Three | 0 | 0 |
| | stage Four | 5 | 0.8 |
| Duration of infection since known | <60 month | 164 | 26.3 |
| | 60–120 month | 189 | 30.3 |
| | ≥120 month | 270 | 43.3 |
| Time to start ART since known infection | ≤1 month | 254 | 40.8 |
| | 2–11 month | 235 | 37.7 |
| | ≥12 month | 134 | 21.5 |
| Base line CD4 | <350 cells/ml | 330 | 53.0 |
| | ≥350 cells/ml | 293 | 47.0 |
| Current Viral Load(VL) status | Unsuppressed | 65 | 10.4 |
| | Suppressed | 558 | 89.6 |
| History of treatment failure | Yes | 118 | 18.9 |
| | No | 505 | 81.1 |
| History of poor adherence | Yes | 217 | 34.8 |
| | No | 406 | 65.2 |

The multivariate analysis in binary logistic regression showed that three out of five variables were significantly associated with the outcome variable. Biological children whose eligible index cases started ART immediately were 85% less likely to be seropositive than those who didn't (AOR = 0.148; 95%CI: 0.067–0.325). Biological children whose index cases' functional status was bedridden or ambulatory at enrollment were 7.71 times more likely to be seropositive compared to those whose status was working (AOR = 7.71; 95%CI: 3.5–17). Biological children whose index case baseline CD4 level was 350 cells/ml were 8.06 times more likely to be seropositive compared to those whose CD4 level was > 350 cells/ml (AOR = 8.06; 95%CI: 1.8–36) (Table 7).

## Factors associated with HIV seropositive status among spousal partners

The bivariate analysis showed sex, wealth index, ever used a condom, had a history or symptoms of STI in the 12 months before the survey among spousal partners and factors from the index cases such as; notification of disclosure status of HIV, ART initiation, if eligible,

**Table 7. Factors associated with seropositive status among biological children of index cases in Sodo Town, 2021 (N = 406).**

| Variables | Category | Serostatus of biological children | | 95%CI | |
|---|---|---|---|---|---|
| | | Positive | Negative | COR | AOR |
| Immediate ART initiation | Yes | 13(3.2%) | 300(73.9%) | 0.083(0.041–0.167) | 0.148(0.067–0.325)*** |
| | No | 32(7.9%) | 61(15.0%) | 1 | 1 |
| Functional status at enrollment | Bedridden or Ambulatory | 32(7.9%) | 51(12.6%) | 14.96(7.36–30.4) | 7.71(3.5–17)*** |
| | Working | 13(3.2%) | 310(76.4%) | 1 | 1 |
| WHO stage at enrollment | Stage1 or 2 | 10(2.5%) | 276(68.0%) | 1 | 1 |
| | Stage3 or 4 | 35(8.6%) | 85(20.9%) | 11.7(5.4–23.9) | 1.53(0.512–4.55) |
| Duration of infection since known | <60 month | 3(0.7%) | 95(23.4%) | 1 | 1 |
| | 60–120 month | 13(3.2%) | 121(29.8%) | 3.4(0.942–12.3) | 0.597(0.119–2.99) |
| | ≥120 month | 29(7.1%) | 145(35.7%) | 6.33(1.88–21.4) | 0.853(0.181–4.02) |
| Base line CD4 | <350 | 43(10.6%) | 167(41.1%) | 24.98(5.96–104) | 8.06(1.8–36)** |
| | ≥350 | 2(0.5%) | 194(47.8%) | 1 | 1 |

* Statistically significant (p = 0.01–<0.05)

** Strong statistical significance (p = 0.001–<0.01)

*** Very strong statistical significance (p< 0.001)

functional status at enrollment, WHO stage at enrollment and duration of infection since known were candidate variables for multivariable binary logistic regression analysis.

In multivariable logistic regression analysis, had a history or symptoms of STI ((AOR = 5.7; 95% CI: 1.86–17.5)), and from the index factors like notification of disclosure of HIV status (AOR = 0.062; 95%CI: 0.024–0.159), ART initiation status(AOR = 0.172; 95%CI: 0.044–0.675), functional status at enrollment, WHO stage at enrollment(AOR = 7.27; 95%CI: 2.37–22.3, and duration of infection since known(AOR = 5.9; 95%CI: 1.8–14.4) were factors significantly associated with HIV seropositive status of spousal partners of index cases.

Spousal partners who had a history or symptoms of STI in the previous 12 months were 5.7 times more likely to be seropositive than their counterparts (AOR = 5.7; 95% CI: 1.86–17.5). Spousal partners whose index cases disclosed their status within a month of diagnosis were 94% less likely to be seropositive than those whose index cases stayed a month or more (AOR = 0.062; CI: 0.024–0.159). Spousal partners whose index cases started ART immediately as eligible were 83% less likely to be seropositive than those whose index cases started immediately (AOR = 0.172; 95%CI: 0.044–0.675). Spousal partners whose index cases were WHO stage III or IV at enrollment were 7.27 times more likely to be seropositive than those whose index cases were WHO stage I or II (AOR = 7.27; 95%CI: 2.37–22.3). Spouses of index cases who had known their status for more than 10 years were 5.9 times more likely to be seropositive than those who had known their status for less than 5 years (AOR = 5.9; 95%CI: 1.8–14.4) (Table 8).

## Discussion

In this study, the HIV seropositive status among family members was 31.5% (95%CI; 27.6–35.2%). This implies that family members of the index cases are at high risk for HIV infection. The prevalence is consistent with the findings of studies conducted at Felege Hiwot Referral Hospital, Northwest Ethiopia and Zimbabwe which was 30% [10] and 32.6% [12] respectively. However, it is lower lower than the prevalence reported from South Africa of 38% [13] and higher than the prevalence reported from Malawi of 22% [14]. The possible explanations for this difference might be due to variation in socio-demographic and economic characteristics, prevention strategies, and implementation differences across the nations.

**Table 8. Factors associated with seropositive status among spousal partners in Sodo Town, 2021 (n = 217).**

| Variables | Category | Serostatus of spousal partner | | 95%CI | |
|---|---|---|---|---|---|
| | | Positive | Negative | COR | AOR |
| Sex | Male | 76(35.0%) | 42(19.4%) | 1 | 1 |
| | Female | 75(34.6%) | 24(11.1%) | 1.73(0.953–3.13) | 1.84(0.75–4.52) |
| Wealth index | Frist percentile | 46(21.2%) | 23(10.6%) | 0.645(0.317–1.31) | 0.551(0.193–1.57) |
| | Second percentile | 43(19.8%) | 23(10.6%) | 0.603(0.295–1.23) | 0.345(0.125–1.949) |
| | Third percentile | 62(28.6%) | 20(9.2%) | 1 | 1 |
| Ever used a condom | Yes | 59(27.2%) | 39(18.0%) | 1 | 1 |
| | No | 92(42.4%) | 27(12.4%) | 2.25(1.25–4.06) | 1.37(0.561–3.36) |
| Had or symptoms of an STI | Yes | 41(18.9%) | 9(4.1%) | 2.36(1.07–5.2) | 5.7(1.86–17.5)** |
| | No | 110(50.7%) | 57(26.3%) | 1 | 1 |
| Notification of Disclosure of index cases | ≥1month | 106(48.8%) | 10(4.6%) | 1 | 1 |
| | <1month | 45(20.7%) | 56(25.8%) | 0.076(0.036–0.162) | 0.062(0.024–0.159)*** |
| Immediate ART initiation of index cases | Yes | 100(46.1%) | 63(29.0%) | 0.093(0.028–0.312) | 0.172(0.044–0.675)* |
| | No | 51(23.5%) | 3(1.4%) | 1 | 1 |
| Functional status at enrollment of index cases | Bedridden or Ambulatory | 40(18.4%) | 2(0.9%) | 11.5(2.7–49.3) | 1.23(0.198–7.68) |
| | Working | 111(51.2%) | 64(29.5%) | 1 | 1 |
| WHO stage at enrollment of index cases | Stage1or 2 | 86(39.6%) | 60(27.6%) | 1 | 1 |
| | Stage3 or 4 | 65(30.0%) | 6(2.8%) | 7.56(3.08–18.6) | 7.27(2.37–22.3)** |
| Duration of infection since known of index cases | <60 month | 31(14.3%) | 35(16.1%) | 1 | 1 |
| | 60–120 month | 40(18.4%) | 15(6.9%) | 3(1.4–6.47) | 3.04(0.994–9.32) |
| | ≥120 month | 80(36.9%) | 16(7.4%) | 5.65(2.74–11.6) | 5.09(1.8–14.4)** |

* Statistically significant (p = 0.01–<0.05)

** Strong statistical significance (p = 0.001–<0.01)

*** Very strong statistical significance (p < 0.001).

In a sub-group-analysis of the prevalence of seropositive status among family members of index cases only on biological children, the prevalence of seropositive status in our study area was found to be 11.1% with a 95%CI of (8.4–14.5%). This finding is consistent with a study done at FelegeHiwot Referral Hospital, Northwest Ethiopia, which found 10.7% [10]. But our finding is much higher compared with other studies, including the Lesotho study that showed a prevalence of 1.4%, Malawi at 4%, and Kenya at 7.4% [15–17]. Possible explanations for this difference might be implementation differences (time of implementation, service coverage) of the prevention of mother-to-child transmission (PMTCT) strategy and socio-demographic and economic differences between nations.

In a sub-group-analysis of the prevalence of seropositive status among family members of index cases only on spousal partners, in our study area, was found to be 69.6% with a 95%CI (63.1–75.6%). This finding is almost similar to a study done at FelegeHiwot Referral Hospital, Northwest Ethiopia; which was 71.5% [10]. But, this finding is lower than in a study conducted in SSA 76.23% [18]. On the other side, this finding is higher than a study conducted in Ethiopia, EDHS, 2016 28% [4]. The possible explanation for this wide gap with the national prevalence might be that participants of index cases were the only new index cases in EDHS, 2016. Likewise, this finding is also higher than the study conducted on 37.7% in Nigeria and 61.9% in Tanzania [19, 20]. The possible explanations for this difference might be ART service coverage, prevention, implementation, and monitoring strategy differences from one country to another country.

This finding revealed that biological children whose index cases started ART immediately, based on eligibility criteria, were 85% less likely to be seropositive than those whose index

cases didn't start. This might be the fact that initiation of ART increases CD4 and inversely decreases the viral load level of index cases, eventually preventing transmission of HIV to their biological children [9].

This study showed that biological children whose index cases were bedridden or ambulatory functional status at enrollment were 7.7 times more likely to be seropositive than those whose status was working. This finding also revealed that biological children whose index case baseline CD4 level was 350 cells/ml were 8.1 times more likely to be seropositive than those whose CD4 level was > 350 cells/ml. There is no study report related to this in this specific age group and almost 90% of biological children acquire the HIV infection through vertical transmission. However, the importance of these factors is emphasized by the national PMTCT guideline of Ethiopia [9]. In index cases that do not start ART as per the eligibility criteria, their CD4 level declines. This leads to viral replication, increase in vulnerabilities to opportunistic infections, and deterioration in patient's functional status, thereby increasing the possibility of vertical viral transmission of HIV. This study revealed that spousal partners who had a history or symptoms of an STI in the 12 months before the survey were 5.7 times more likely to be seropositive than their counterparts. It is consistent with the study conducted in 30 provinces of China [21]. The possible explanation might be that since STI and HIV have a linear relationship, and STI may be characterized by genital sores or ulcers, inguinal swelling that is exposed to HIV transmission. This finding is supported by EDHS, 2016 [4].

This finding of this study shows that spousal partners whose index cases disclosed their status within a month of diagnosis were 94% less likely to be seropositive than those whose index cases who stayed for a month or more after diagnosis. This implies that disclosure of their HIV positive status to their spousal partners raises awareness about their risk and need for testing. Disclosure also has important health benefits. It increases social support, fosters closer relationship to others, increases testing uptake, creates good prevention opportunities like condom utilization, and other prevention methods, and improves treatment adherence and retention, which increases CD4 and suppresses viral load level that leads to halting of transmission. This finding is supported by the National Comprehensive HIV Care Guideline of Ethiopia, 2018 [8].

This finding revealed that those spousal partners whose index cases started ART immediately as on meeting the eligibility criteria criteria were 83% less likely to be seropositive than those whose index partners didn't start immediately. This finding is consistent with the findings of a study conducted in Addis Ababa, Ethiopia [22]. This might be because ART initiation based on the eligibility criteria could lead to increase in index case CD4 count and VL suppression that decreases HIV transmission to spousal partners This study showed that spousal partners whose index cases who were at WHO clinical stage III or IV at enrollment were 7.25 times more likely to be seropositive than spousal partners whose index cases who were WHO clinical stage I or II. This finding is with the study conducted in Thailand and North Vietnam [23, 24]. This could be due to the fact that index cases who were at WHO stage III or IV had severe opportunistic infections that exposed the index case to decreased their CD4 levels and increased the viral load levels resulting in high viral transmission to spousal partners.

This study also revealed that spousal partners whose index cases lived with the virus since they knew their status for more than 10 years were 5.09 times more likely to be seropositive than those who knew their status for less than 5 years. It is consistent with the study conducted in Thailand and Rakia, Uganda [23, 25]. This implies that as the duration of the relationship between spousal partners increases, the rate of HIV transmission also increases.

### Strength and limitation

Because the factors of biological children and spousal partners of index cases are different, being able to address both outcomes in one study was the strength of this study. Family members whose index cases did not disclosed their status or had follow-up outside of Sodo Town were not included in the study. Obstetric and as well as modes of transmission other than MTCT were excluded as the study focused on biological factors

### Conclusion

In this study, HIV among family members of index cases in general and specifically in both biological children and spousal partners in Sodo Town is high. Late ART initiation as per the eligibility criteria, bedridden or ambulatory functional status during enrollment, and baseline CD4 350 cells/ml of index cases were the major factors associated with HIV seropositive status. in the spousal partners with HIV history or symptoms of STI, and whereas among the index cases, late notification of disclosure status, late ART initiation as per the eligibility criteria, WHO stage III or IV at enrollment, and duration of HIV infection since known of $> 10$ years were factors for the seropositive. In the context of Sodo Town, comprehensive HIV prevention, care, and treatment services need be promoted and implemented among family members of the index cases to increase the achievements of program targets in HIV epidemic control.

### Supporting information

**S1 Data.**
(SAV)

### Acknowledgments

We are grateful to the Wolaita Sodo University, College of Medicine and Health Sciences for providing ethical clearance for this study. We are also indebted to our data collectors, supervisors, and the study participants.

### Author Contributions

**Conceptualization:** Alemayehu Kefale, Kassa Daka.

**Data curation:** Alemayehu Kefale, Kassa Daka, Amene Abebe, Dereje Haile, Abdulbasit Sherfa.

**Formal analysis:** Alemayehu Kefale, Kassa Daka, Amene Abebe, Dereje Haile, Kebreab Paulos, Abdulbasit Sherfa.

**Funding acquisition:** Alemayehu Kefale, Dereje Haile, Animut Addis.

**Investigation:** Alemayehu Kefale, Kassa Daka, Amene Abebe, Dereje Haile, Kebreab Paulos, Abdulbasit Sherfa, Animut Addis, Asaminew Ayza, Jegnaw Wolde.

**Methodology:** Alemayehu Kefale, Kassa Daka, Amene Abebe, Dereje Haile, Kebreab Paulos, Abdulbasit Sherfa, Animut Addis, Muluken Gunta, Asaminew Ayza, Jegnaw Wolde.

**Project administration:** Alemayehu Kefale, Amene Abebe, Kebreab Paulos, Animut Addis, Muluken Gunta, Asaminew Ayza, Jegnaw Wolde.

**Resources:** Alemayehu Kefale, Dereje Haile, Animut Addis, Asaminew Ayza, Jegnaw Wolde.

**Software:** Alemayehu Kefale, Kassa Daka, Amene Abebe, Dereje Haile, Kebreab Paulos, Asaminew Ayza.

**Supervision:** Alemayehu Kefale, Amene Abebe, Dereje Haile, Abdulbasit Sherfa, Animut Addis, Muluken Gunta, Asaminew Ayza, Jegnaw Wolde.

**Validation:** Alemayehu Kefale, Kassa Daka, Amene Abebe, Dereje Haile, Asaminew Ayza.

**Visualization:** Alemayehu Kefale, Kassa Daka, Amene Abebe, Dereje Haile, Asaminew Ayza.

**Writing – original draft:** Alemayehu Kefale, Kassa Daka, Dereje Haile, Kebreab Paulos.

**Writing – review & editing:** Alemayehu Kefale, Kassa Daka, Amene Abebe, Dereje Haile, Kebreab Paulos, Abdulbasit Sherfa, Animut Addis, Muluken Gunta, Asaminew Ayza, Jegnaw Wolde.

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
