## [Decision Letter · Decision Letter 0]

31 Aug 2022

PONE-D-22-04620PREVALENCE OF HIV SEROPOSITIVESTATUS AND ASSOCIATED FACTORS AMONG FAMILY MEMBERS OF INDEX CASES OF ART CLINICATTENDANTS INSODO TOWN, SOUTHERN ETHIOPIAPLOS ONE

Dear Dr. Haile,

Thank you for submitting your manuscript to PLOS ONE. After careful consideration, we feel that it has merit but does not fully meet PLOS ONE’s publication criteria as it currently stands. Therefore, we invite you to submit a revised version of the manuscript that addresses the points raised during the review process.

Specifically although the study was found to be well conducted and the statistical analysis reasonable, extensive re-wrting of the grammar and reorganization of the manuscript are needed to make the manuscript easier to read and understand. Included is a list of suggested changes to improve the written english of the manuscript.

We look forward to receiving your revised manuscript.

Kind regards,

Colin Johnson, Ph.D.

Academic Editor

PLOS ONE

Journal Requirements:

2. Thank you for indicating in your ethics statement that informed written and verbal consent was obtained from each spousal partner and assent for biological children. Based on this statement it is not clear whether the informed consent was obtained from the participants included in the study. Please revise your statement to reflect this.

Reviewers' comments:

Reviewer's Responses to Questions

**Comments to the Author**

1. Is the manuscript technically sound, and do the data support the conclusions?

Reviewer #1: Yes

2. Has the statistical analysis been performed appropriately and rigorously? 

Reviewer #1: Yes

3. Have the authors made all data underlying the findings in their manuscript fully available?

Reviewer #1: Yes

4. Is the manuscript presented in an intelligible fashion and written in standard English?

Reviewer #1: No

5. Review Comments to the Author

Reviewer #1: The Manuscript required extensive reviewing, particularly in terms of language (literally requiring a line by by line reviewing). The formatting lacks the very basics like line spacing and text alignments. Seems that this manuscript slipped though the standard PLoS pre-reviewing. As my comments are, out of necessity, more than 20,000 characters, I have included the file as an attachment.

6. PLOS authors have the option to publish the peer review history of their article (what does this mean?). If published, this will include your full peer review and any attached files.

Reviewer #1: **Yes: **Sileshi Lulseged

---

## [Author Response · Author response to Decision Letter 0]

20 Oct 2022

Manuscript #: PONE-D-22-04620 

Title: Seropositive Status and Associated Factors among Family Members of Index Cases of ART clinic Attendants in Sodo Town, Southern Ethiopia

Comments to the Author

1. Is the manuscript technically sound, and do the data support the conclusions?

Reviewer #1: Yes

2. Has the statistical analysis been performed appropriately and rigorously?

Reviewer #1: Yes

3. Have the authors made all data underlying the findings in their manuscript fully available?

Reviewer #1: Yes

4. Is the manuscript presented in an intelligible fashion and written in standard English?

Reviewer #1: No

 

5. Review Comments to the Author

General Comments:

focused objective and has generated the required data and performed an appropriate analysis of the data and interpretation of the results. It has generated conclusions warranted by the data. However, the language of the submitted manuscript for from adequate, both from the perspective of clearly conveying what is intended and quality standards of scientific publications. Here, review comments are provided that the authors may wish to consider improving the quality of the manuscript, particularly in terms of content organization, formatting, and language to make the manuscript potentially publishable in PLOS.

Authors need to use the ‘PLoS Guidelines to Authors’ in the formatting (line spacing, text alignment) as well as in organizing the content in the different sections of the manuscript. Please note that some of the comments given in the different sections below clearly indicate this.

The manuscript requires heavy language editing – authors may wish to enlist support from a language editor. Much language editing in the manuscript is done as part of this review (please see below comments in each section) 

Thank you for your patience and meticulous observation of this manuscript and your point-by-point comments. We have carefully addressed the comments raised accordingly.

Specific comments:

Title/Title Page

The title should be written in sentence case (capitalize only the first word of the title and proper names); expand ART to Antiretroviral Treatment

Response: thank you for your comment; we have corrected and highlighted it in the revised manuscript

Line 1: Missing space between Seropositiveand Status 

Response: thank you for your comment; we have corrected and highlighted it in the revised manuscript

Line 3: Missing space between words. Correct CLINICATTENDANTSINSODO to Clinic Attendants in Sodo

Response: thank you for your comment; we have corrected and highlighted it in the revised manuscript

Line 4: Would edit Abebe 2Dereje asread Abebe2, Dereje

Response: thank you for your comment; we have corrected and highlighted it in the revised manuscript

Line 5: Would remove the extra comma between Addis4 and Muluken

Response: thank you for your comment; we have corrected and highlighted it in the revised manuscript

Lines 6-9: Would edit the affiliation for each author (include department, University, or organization affiliation, and its location, including city, state/province, and country

Response: thank you for your comment; we have corrected and highlighted it in the revised manuscript

Would add on this page: first author’s e-mail address followed by name initials in parenthesis

Response: thank you for your comment; we have corrected and highlighted it in the revised manuscript

On this page: also indicate authors’ contribution(s) using symbols as given in PLoS One “Title, Author, Affiliation Formatting Guidelines”.

Abstract:

Lines 30-31: the sentence needs editing; suggest "... transmitted through … sexual contact with an infected partner and babies born to mothers infected with the virus.”

Response: thank you for your comment; we have corrected and highlighted it in the revised manuscript

Lines 33-35: The sentence is the title repeated as is; suggest restating this in a form of an objective ... Like “This study aimed at identifying prevalence and determinants of HIV infection among family members of index cases on antiretroviral treatment (ART).”

Response: thank you for your comment; we have corrected and highlighted it in the revised manuscript

Line 38: Would Insert HIV between “of” and “index”

Response: thank you for your comment; we have corrected and highlighted it in the revised manuscript

Line 39: “.. collected data ……” on what ?

Response: thank you for your comment; we have corrected and highlighted it in the revised manuscript

Line 39: Would add space after the stop after data. 

Response: thank you for your comment; we have corrected and highlighted it in the revised manuscript

Line39: Would replace “investigate” with “identify”

Response: thank you for your comment; we have corrected and highlighted it in the revised manuscript

Line40: Helpful to add the expanded form for AOR upfront here so that we use the acronym in the rest f the manuscript; i.e., adjusted odds ratio (AOR)

Response: thank you for your comment; we have corrected and highlighted it in the revised manuscript

Line41: Same here. 95% Confidence interval (CI) 

Response: thank you for your comment; we have corrected and highlighted it in the revised manuscript

Lines41: The P in P-values is written in upper case as is and italicised (P). Please do this across the manuscript.

Response: thank you for your comment; we have corrected and highlighted it in the revised manuscript

Lines41-42: last part of the sentence requires editing. Suggest changing “----used to declare the cut-off point in determining the level of significance” to “---- used as a cut-off point to determine the level of statistical significance of point estimate.” 

Response: thank you for your comment; we have corrected and highlighted it in the revised manuscript

Line 45:Would edit“----- revealed that 31.5% with 95%CI (27.6-35.2%)’ as “----- revealed that 31.5% (95%CI: 27.6-35.2%); need to use same formant across the manuscript.

Response: thank you for your comment; we have corrected and highlighted it in the revised manuscript.

Lines 46-47: Would edit:(…11.1% with 95%CI (8.4–14.5%..) as (…11.1% (95%CI: 8.4–14.5%)

Response: thank you for your comment; we have corrected and highlighted it in the revised manuscript

Line 47:Would edit:“…and 69.6% with 95%CI (63.1–75.6%..)” as“….. and 69.6% (95%CI: 63.1–75.6%)

Response: thank you for your comment; we have corrected and highlighted it in the revised manuscript

Line 50:Would insert ‘of’ between “level” and “350”

Response: thank you for your comment; we have corrected and highlighted it in the revised manuscript

Line 51:Would edit “….. significant with ….” as “….. significantly associated with …”

Response: thank you for your comment; we have corrected and highlighted it in the revised manuscript

Line 54:Would edit:“…. infection since known …”, may remove ‘since known’ as this does not add an y useful information.

Response: thank you for your comment; we have corrected and highlighted it in the revised manuscript

Line 59:Would change ‘should be’ to 'might be' or 'need to'; one needs to avoid strong words like should in conclusions/recommendations.

Response: thank you for your comment; we have corrected and highlighted it in the revised manuscript.

Introduction

Line 60: May consider editing as “HIV infection, HIV seropositive, biological children, spousal partners, index case,”.

Response: thank you for your comment; we have corrected and highlighted it in the revised manuscript.

Line 63: Definition is incomplete, and the language requires editing. Option: "Index testing is a voluntary process where counsellors or health care providers ask a newly diagnosed HIV positive patient or an HIV positive patient already accessing HIV treatment to list all partners who could have been exposed to HIV through them, the index cases" [1].

Response: thank you for your comment; we have corrected and highlighted it in the revised manuscript.

Line 57: Just use the acronym ‘ICT’; already used at first enter above with the expanded forms.

Response: thank you for your comment; we have corrected and highlighted it in the revised manuscript.

Line 67: Insert space “ ….models of testing:(1)Facility…..” so that it reads“ ….models of testing (1) Facility…..”

Line:67:Would edit “… (1)Facility Based ICT …” as “ ……. (1) Facility-based ICT …”

Response: thank you for your comment; we have corrected and highlighted it in the revised manuscript.

Line 68; Would edit “…. index case and gets the names ..” to read “…. index cases on ART at the health facility, and gets the names .. “ 

Response: thank you for your comment; we have corrected and highlighted it in the revised manuscript.

Line 69: Would replace ‘old’ with “of age’

Response: thank you for your comment; we have corrected and highlighted it in the revised manuscript.

Line 69: Would Insert a comma after ‘old’ and delete “in Antiretroviral Treatment (ART) service giving facility. 

Response: thank you for your comment; we have corrected and highlighted it in the revised manuscript.

Line 69-72: Sentence after ‘(2)’ needs revision, as it has many grammatic errors; suggest: “community-based ICT: a counsellor traces sexual partner(s) of index cases and their biological children who are under 15 years of age through home-to-home visits, offers HTS, and links newly identified People Living with HIV (PLHIV) to ART services [1].

Response: thank you for your comment; we have corrected and highlighted it in the revised manuscript.

Line 75. Would add ‘year’ at the end of the sentence.

Response: thank you for your comment; we have corrected and highlighted it in the revised manuscript.

Line 76. Paragraph starting with this line is a continuation of the above; Would keep this para with the above.

Response: thank you for your comment; we have corrected and highlighted it in the revised manuscript.

Line 76: Would edit “….. PLHIV were lived and …” as “…. PLHIV were living in 2021, and of which ….”

Response: thank you for your comment; we have corrected and highlighted it in the revised manuscript.

Line 77: In “… transformative and ---” would add ‘a’ before transformative.

Response: thank you for your comment; we have corrected and highlighted it in the revised manuscript.

Line 78: Would edit “ …. expansion of ART indicating that from total of ….”as “…… as indicated by the total number of …”

Response: thank you for your comment; we have corrected and highlighted it in the revised manuscript.

Line 79: Would edit “…. sixteen million (16 million) …” as “..16 million…”

Response: thank you for your comment; we have corrected and highlighted it in the revised manuscript.

Line 79: Would need space between treatment and the citation in“…. on treatment[2] ….”.Would do this consistently all along.

Response: thank you for your comment; we have corrected and highlighted it in the revised manuscript.

Line 80: Would replace ”…among which …” by “… of which ….”

Response: thank you for your comment; we have corrected and highlighted it in the revised manuscript.

Line 82: Would keep this paragraph starting with line with the one above paragraph.

Response: thank you for your comment; we have corrected and highlighted it in the revised manuscript.

Line 82: Would delete ‘also”

Response: thank you for your comment; we have corrected and highlighted it in the revised manuscript.

Line 82.Would replace …among which….” With “… of which …”

Response: thank you for your comment; we have corrected and highlighted it in the revised manuscript.

Line 83: Would edit “… of age, with 13,000 …” to read “ of age. There were 13,000….”

Response: thank you for your comment; we have corrected and highlighted it in the revised manuscript.

Line 83: Would edit “ …. under 15 years old.” as “….. under 15 years of age. ….”

Response: thank you for your comment; we have corrected and highlighted it in the revised manuscript.

Line 84: Would edit “PLHIVs” should be “PLHIV”

Response: thank you for your comment; we have corrected and highlighted it in the revised manuscript.

Line 86: Would merge the paragraph starting with “according to the Ethiopian ….” with the above paragraph

Response: thank you for your comment; we have corrected and highlighted it in the revised manuscript.

Line 87: Would edit “….was 0.9 percent (0.7 percent ….” as “… was 0.9% (0.7% …..”; and be consistent all along.

Response: thank you for your comment; we have corrected and highlighted it in the revised manuscript.

Line 88: Would replace “With this general prevalence, the current epidemiology of HIV is heterogeneous, with ….” with 'HIV/AIDS in Ethiopia is regularly categorized as “generalized“ among the adult population, with …….”

Response: thank you for your comment; we have corrected and highlighted it in the revised manuscript.

Line 90: Would replace “…. Geographical ….” with ““…. Geographic ….”

Response: thank you for your comment; we have corrected and highlighted it in the revised manuscript.

Line 90: Would insert a reference (e.g., the Ethiopian Population-based HIV Impact Assessment 2020) at end of the line (after subgroups)

Response: thank you for your comment; we have added and highlighted it in the revised manuscript.

Line 92: Would insert after “below 0.1” a reference (e.g., EPHI, Mini EDHS 2018/19).

Response: thank you for your comment; we have corrected and highlighted it in the revised manuscript.

Line 92: Would replace ‘Somalia” with ‘Ethiopian Somali Region”

Response: thank you for your comment; we have corrected and highlighted it in the revised manuscript.

Line 97:Paragraph starting with this is redundant, repeating what is covered in the above paragraph. Would delete.

Response: thank you for your comment; we have deleted and highlighted it in the revised manuscript.

Line 101: Would insert HIV between universal and testing and edit “ …. testing strategy …”, as“…..test and treat strategy ….”

Response: thank you for your comment; we have corrected and highlighted it in the revised manuscript.

Line 101: Would edit “…. testing all individuals as a mass ……” as“ …. testing all individuals in the population who were at risk …”

Response: thank you for your comment; we have edited and highlighted it in the revised manuscript.

Line 102: Would edit :…. index case clients are one of these areas [6] …” as“….index cases as a target population for this strategy [6].

Response: thank you for your comment; we have edited and highlighted it in the revised manuscript.

Line 104: Would keep paragraph starting with this line with the above one.

Response: thank you for your comment; we have corrected and highlighted it in the revised manuscript.

Line 105: Would edit “….and Family-Based Index Case Testing ….” as“…..Family-based Index Case Testing ……”

Response: thank you for your comment; we have corrected and highlighted it in the revised manuscript.

Line 105: Would edit “……. or Index Case Testing (ICT) strategy ….” as “……. or the ICT strategy ….”

Response: thank you for your comment; we have corrected and highlighted it in the revised manuscript.

Line 107: Would edit “…..under-15-year old…” to“…..under 15-year-old…”

Response: thank you for your comment; we have corrected and highlighted it in the revised manuscript.

Line 107: Would edit: “…… biological children to break the chain of ……. “ as “…… biological children. This breaks the chain of ……. “ NB. This breaks a long sentence.

Response: thank you for your comment; we have corrected and highlighted it in the revised manuscript.

Line 108: Would edit: “….. who have been exposed to HIV …”as“….. who are exposed to HIV …”

Response: thank you for your comment; we have corrected and highlighted it in the revised manuscript.

Line 108: Would edit “….. to HIV to give preventive …..” as “to HIV, provide preventive ….”

Response: thank you for your comment; we have corrected and highlighted it in the revised manuscript.

Line 109: Would edit “…… services for negative and, ….” as“…… services for those testing HIV-negative, and, ….

Response: thank you for your comment; we have corrected and highlighted it in the revised manuscript.

Line 109: Would edit “…. if positive, linkage and treatment that increase case detection and attain ….” as “. if positive, link them to cared and treatment services, a cascade that will contribute to attaining the..”

Response: thank you for your comment; we have corrected and highlighted it in the revised manuscript.

Lines 110-111: Would edit “…. while family members have an ongoing risk of contracting HIV.” as“….. exposing the family members to an ongoing risk of contracting HIV.”

Response: thank you for your comment; we have corrected and highlighted it in the revised manuscript.

Line 111:Would replace “So” with “Therefore”

Response: thank you for your comment; we have corrected and highlighted it in the revised manuscript.

Lines 112-113: Would edit “….under 15 years of biological children[1].” as “….under 15-years old biological children[1].”

Response: thank you for your comment; we have corrected and highlighted it in the revised manuscript.

Line 117: The paragraph starting with this line would rather be kept with the above one.

Response: thank you for your comment; we have corrected and highlighted it in the revised manuscript.

Line 120-122: Would replace “Testing for HIV and linking to care for those identified by the families of the index cases will improve the health outcomes of the seropositive family members[5].” with “Testing for HIV and linking those who are positive to care and treatment services will improve the health outcomes of the seropositive family members[5].

Response: thank you for your comment; we have corrected and highlighted it in the revised manuscript.

“Line 123: Would replace ‘But’ with :Nevertheless’

Response: thank you for your comment; we have corrected and highlighted it in the revised manuscript.

Line 123: Would edit “HIV positive” as “HIV-positive”; and do this consistently across the manuscript

Response: thank you for your comment; we have corrected and highlighted it in the revised manuscript.

Lines 123-127: “…people for HIV testing and associated factors among ART clinic attendants at FelegeHiwot Referral Hospital, Northwest Ethiopia, reveals that 25.8% of index cases were not tested. Even though there were a significant number of family members not tested in this study, 30% of family members were seropositive[8].” requires editing; Would consider “……. ART clinic attendants at a referral hospital in Northwest Ethiopia revealed that many children and adults were traced and tested for HIV. Yet, there were many untested partners, children and other individuals living with index cases who were not tested.

Response: thank you for your comment; we have edited and highlighted it in the revised manuscript.

Lines 128-130: Would delete this para; already addressed in the paragraphs above.

Response: thank you for your comment; we have deleted and highlighted it in the revised manuscript.

Line131: Paragraph starting with this line would rather be kept with paragraph above ending with ref #8.

Response: thank you for your comment; we have edited and highlighted it in the revised manuscript.

Line 131: Would edit “…. no study was conducted …” as“… no similar study was conducted …”

Response: thank you for your comment; we have edited and highlighted it in the revised manuscript.

 ‘Line 134: Would replace ‘’would’ with “will”

Response: thank you for your comment; we have replaced and highlighted it in the revised manuscript.

Lines 134-136: “ ….. the family members' information in routine clinical practice and could facilitate ending the HIV/AIDS epidemic vision in 2030.” requires editing; would suggest: :… 'HIV serostatus information on family members of index cases, which will contribute to improving HIV care and testament services and epidemic control in Ethiopia'. 

Response: thank you for your comment; we have edited and highlighted it in the revised manuscript.

Methods

Line 138: Usually written as ‘Materials and Methods” or just “Methods”

Response: thank you for your comment; we have edited and highlighted it in the revised manuscript.

Line 142: Would expand SNNPR. Southern Nations, Nationalities, and Peoples Region (SNNPR).

Response: thank you for your comment; we have expanded and highlighted it in the revised manuscript.

Line 142: Would edit “… Sodo Town has been…” as“… Sodo Town is …”

Response: thank you for your comment; we have edited and highlighted it in the revised manuscript.

‘Line 143: Would replace ‘3’ with ‘three’

Response: thank you for your comment; we have replaced and highlighted it in the revised manuscript.

Line 143: Would edit as “ … kebele (lowest administrative entity)”

Response: thank you for your comment; we have edited and highlighted it in the revised manuscript.

Line 144: Would insert reference for “According to the 2021 population and housing census projection [ ]. …”

Response: thank you for your comment; we have edited and highlighted it in the revised manuscript.

Line 146: Would replace ‘3’ with ‘three’

Response: thank you for your comment; we have edited and highlighted it in the revised manuscript.

Line 146: Would edit: “.. service-giving facilities ….” as“.. service-delivery facilities ….”

Response: thank you for your comment; we have edited and highlighted it in the revised manuscript.

Line 147: Would replace ‘2” with ‘two” and ‘1’ with ‘one’

Response: thank you for your comment; we have edited and highlighted it in the revised manuscript.

Line 150: Would replace sentence with “The study was a community-based cross-sectional study”

Response: thank you for your comment; we have edited and highlighted it in the revised manuscript.

Line 15

---

## [Decision Letter · Decision Letter 1]

17 Nov 2022

PONE-D-22-04620R1Prevalence Of HIV Seropositive  Status  and Associated  Factors Among Family Members Of Index Cases Of Antiretroviral Clinical  Attendants in Sodo Town, Southern EthiopiaPLOS ONE

Dear Dr. Haile,

Thank you for submitting your manuscript to PLOS ONE. After careful consideration by a referee, a list of

grammar suggestions that may improve the manuscript were provided. Please examine the list of possible changes and submit a revised manuscript. Please submit your revised manuscript by Jan 01 2023 11:59PM. If you will need more time than this to complete your revisions, please reply to this message or contact the journal office at plosone@plos.org. Please include the following items when submitting your revised manuscript:A rebuttal letter that responds to each point raised by the academic editor and reviewer(s). You should upload this letter as a separate file labeled 'Response to Reviewers'.A marked-up copy of your manuscript that highlights changes made to the original version. You should upload this as a separate file labeled 'Revised Manuscript with Track Changes'.An unmarked version of your revised paper without tracked changes. You should upload this as a separate file labeled 'Manuscript'.If applicable, we recommend that you deposit your laboratory protocols in protocols.io to enhance the reproducibility of your results. Protocols.io assigns your protocol its own identifier (DOI) so that it can be cited independently in the future. For instructions see: https://journals.plos.org/plosone/s/submission-guidelines#loc-laboratory-protocols. Additionally, PLOS ONE offers an option for publishing peer-reviewed Lab Protocol articles, which describe protocols hosted on protocols.io. Read more information on sharing protocols at https://plos.org/protocols?utm_medium=editorial-email&utm_source=authorletters&utm_campaign=protocols.

We look forward to receiving your revised manuscript.

Kind regards,

Colin Johnson, Ph.D.

Academic Editor

PLOS ONE

Journal Requirements:

Reviewers' comments:

Reviewer's Responses to Questions

**Comments to the Author**

1. If the authors have adequately addressed your comments raised in a previous round of review and you feel that this manuscript is now acceptable for publication, you may indicate that here to bypass the “Comments to the Author” section, enter your conflict of interest statement in the “Confidential to Editor” section, and submit your "Accept" recommendation.

Reviewer #1: (No Response)

2. Is the manuscript technically sound, and do the data support the conclusions?

Reviewer #1: Yes

3. Has the statistical analysis been performed appropriately and rigorously? 

Reviewer #1: Yes

4. Have the authors made all data underlying the findings in their manuscript fully available?

Reviewer #1: Yes

5. Is the manuscript presented in an intelligible fashion and written in standard English?

Reviewer #1: No

6. Review Comments to the Author

Reviewer #1: There are grammatic corrections that need to made by authors and section that need to be revised (e,g. references). These are highlighted in the revised manuscript with sticky notes - uploaded as an attachment.

7. PLOS authors have the option to publish the peer review history of their article (what does this mean?). If published, this will include your full peer review and any attached files.

Reviewer #1: **Yes: **Sileshi Lulseged

---

## [Author Response · Author response to Decision Letter 1]

5 Dec 2022

Reviewer #1: There are grammatic corrections that need to made by authors and section that need to be revised (e,g. references). These are highlighted in the revised manuscript with sticky notes - uploaded as an attachment.

Response: Thank you for your comment. We have corrected and highlighted it in the revised manuscript.

---

## [Editor Report · Decision Letter 2]

20 Dec 2022

PONE-D-22-04620R2Prevalence Of HIV Seropositive  Status  and Associated  Factors Among Family Members Of Index Cases Of Antiretroviral Clinical  Attendants in Sodo Town, Southern EthiopiaPLOS ONE

Dear Dr. Haile,

Thank you for submitting your manuscript to PLOS ONE. The manuscript you submitted is significantly improved however the Reviewer has raised a question with regard to calculating the smallest sample size for the size needed to show the effect. This methods and analysis question as well as some minor suggestions are attached a document. If you could provide a revised manuscript that addresses this question, and  make any other necessary changes, I anticipate moving forward with accepting the study.

We look forward to receiving your revised manuscript.

Kind regards,

Colin Johnson, Ph.D.

Academic Editor

PLOS ONE
---

## [Author Response · Author response to Decision Letter 2]

29 Dec 2022

Response to the reviewer: thank you for your meticulous observation and comment. In this study, a sample size was determined using the single population proportion formula by assuming a proportion of 50%, a margin of error of 5%, a 95% confidence interval, and a non-response rate of 10%. The calculated sample size was 423. However, a total of 651 eligible index cases of HIV were identified from the registry, so we used the census method to recruit all study participants.

---

## [Editor Report · Decision Letter 3]

4 Jan 2023

Prevalence Of HIV Seropositive  Status  and Associated  Factors Among Family Members Of Index Cases Of Antiretroviral Clinical  Attendants in Sodo Town, Southern Ethiopia

PONE-D-22-04620R3

Dear Dr. Haile,

We’re pleased to inform you that your manuscript has been judged scientifically suitable for publication and will be formally accepted for publication once it meets all outstanding technical requirements.

Kind regards,

Colin Johnson, Ph.D.

Academic Editor

PLOS ONE
---

## [Editor Report · Acceptance letter]

1 Feb 2023

PONE-D-22-04620R3 

Prevalence of HIV Seropositive Status and Associated Factors among Family Members Of Index Cases Of Antiretroviral Clinical Attendants in Sodo Town, Southern Ethiopia 

Dear Dr. Haile:

I'm pleased to inform you that your manuscript has been deemed suitable for publication in PLOS ONE. Congratulations! Your manuscript is now with our production department. 

Kind regards, 

on behalf of

Dr. Colin Johnson 

Academic Editor

PLOS ONE